# ReCycle Net: Cycle-Aware, Feature-Free GNN for Community Detection

**Caleb Fernandes**
University of South Florida
Tampa, FL, USA
calebf23@usf.edu

**Behnaz Moradi Jamei**
James Madison University
Harrisonburg, VA, USA
moradibx@jmu.edu

## Abstract

Community detection is a fundamental problem in network science, yet classical methods suffer from resolution limits and limited adaptability, while many Graph Neural Networks (GNNs) do not explicitly model higher-order cyclic structures that underlie real-world communities. We propose *ReCycle Net (RCN)*, a feature-free, cycle-aware GNN that integrates Renewal Non-Backtracking Random Walk (RN-BRW) reinforcement into a GAT-style backbone and is trained with a multi-term unsupervised objective combining modularity, Laplacian smoothness, contrastive consistency, and orthogonality regularization. RCN targets graphs where cyclic closure is structurally informative for community formation and learns embeddings that support unsupervised community recovery. Across standard benchmarks, RCN is competitive with strong baselines (e.g., PolBooks: NMI $0.60$, ARI $0.67$; Facebook: silhouette $0.85$), with its clearest gains appearing on overlapping protein complexes under overlap-aware evaluation (Complex Portal: ONMI $0.344$ at $r = 2$ vs. $0.243$, $0.232$, and $0.140$ for generative overlapping-based, strong attention-based, and modularity-based baselines). On graphs with weaker cyclic closure (e.g., Cora), gains are smaller and RCN remains comparable to standard baselines. Overall, these results suggest that explicitly incorporating cycle-derived structure into GNN learning can be beneficial in cycle-rich regimes while remaining robust outside this setting.

## 1 Introduction

Community detection (CD) is a foundational problem in network science, aiming to partition a graph into groups of nodes (communities) that are densely connected internally while sparsely connected externally. In many real-world settings, these groups correspond to meaningful latent structure—for example, social circles in friendship graphs or molecular complexes in protein–protein interaction (PPI) networks. Despite decades of study, accurate CD remains difficult at scale because real graphs are sparse, noisy, and heterogeneous, and because community structure is often non-disjoint (overlapping), multi-scale, or domain dependent. Classical methods such as modularity maximization (e.g., Louvain) and spectral clustering provide strong baselines, but they are limited by resolution effects, sensitivity to noise, and the absence of learnable mechanisms that can adapt to domain-specific structural cues.

A particularly important cue in many networks is *cyclic structure*. Cycles encode higher-order dependencies beyond pairwise connectivity: triangles, feedback loops, and short cyclic motifs frequently mark tightly coupled functional units. This is especially salient in PPI graphs, where protein complexes form densely interdependent interaction patterns and closure through short cycles is common. Motivated by this, Moradi-Jamei et al. (2021) introduced the *Renewal Non-Backtracking Random Walk* (RNBRW), which highlights cyclically important regions by terminating random walks upon detecting a cycle and using retracing statistics to assign cycle-informed edge reinforcement weights. RNBRW provides a scalable, topology-driven signal that can improve downstream clustering when cyclic motifs are structurally meaningful. However, RNBRW by itself is not a learnable model and cannot directly leverage modern representation learning objectives.

Graph Neural Networks (GNNs) have emerged as powerful tools for representation learning on networks. Yet most existing architectures rely on local neighborhood aggregation and message passing, and do not explicitly treat cycles as first-class structural signals. This motivates the question: *can cycle information be integrated into a learnable, unsupervised framework to improve community recovery on cycle-rich graphs such as PPI networks*. Most community detection-oriented GNN pipelines do not explicitly treat cycle structure as a first-class signal. **To address this, we propose the *ReCycle Net (RCN)***, a feature-free, cycle-aware GNN for community detection. RCN integrates RNBRW-derived reinforcement into a GAT-style backbone and trains embeddings using a multi-term unsupervised objective that combines modularity-style structure, Laplacian smoothness, contrastive consistency, and an orthogonality regularizer. Communities are then extracted from the learned embeddings via clustering, and—for overlapping ground truth such as Complex Portal—we evaluate using overlap-aware ONMI with a simple top-$r$ membership rule. Empirically, we show that overlap-aware evaluation is necessary to fairly assess PPI complex recovery and that RCN substantially improves overlap ONMI over strong baselines on Complex Portal.

## 2 RELATED WORK

Community detection aims to identify groups of nodes that are more densely connected internally than externally, often corresponding to functional or semantic units in social, biological, and information networks. Classical approaches such as Louvain-style modularity maximization and spectral clustering optimize global structural objectives (e.g., modularity, graph cuts, or Laplacian eigenstructure). While strong baselines, these methods are non-learnable and can be sensitive to noise, resolution effects, and domain-specific structure such as overlap or multi-scale communities, and they inherently assume disjoint partitions.

Graph Neural Networks (GNNs) provide learnable frameworks that map nodes to embeddings via message passing, enabling unsupervised or self-supervised clustering. Foundational architectures such as Graph Convolutional Networks (GCN; (Kipf and Welling, 2017)) and Graph Attention Networks (GAT; (Veličković et al., 2018)) underlie many modern clustering pipelines. GCNs aggregate normalized neighborhood information, while GATs learn attention weights to prioritize neighbors. However, these architectures primarily operate on local neighborhoods and typically do not treat cyclic motifs (triangles, short loops, feedback patterns) as explicit structural signals, even though such motifs can be informative for community structure, particularly in cycle-rich graphs such as PPI networks, where short cycles often correspond to stable protein complexes.

A growing literature develops unsupervised GNN-based clustering objectives. DMoN (Tsitsulin et al., 2023) performs deep modularity optimization with orthogonality constraints and is a strong modularity-preserving baseline. Related directions include graph autoencoder-based clustering (e.g., DAEGC (Wang et al., 2019), SDCN (Bo et al., 2020)) and self-supervised or contrastive representation learning (e.g., BGRL (Thakoor et al., 2023), GRACE (Xu et al., 2021), DGI (Yin et al., 2022)). A widely used baseline for overlapping community detection is BigCLAM (Yang and Leskovec, 2013), which learns community affiliation strengths for each node under a generative edge formation model, allowing nodes to belong to multiple communities simultaneously. Many of these methods either assume informative node attributes, focus on pairwise neighborhood similarity, or incorporate higher-order structure only implicitly through multi-hop aggregation, which can dilute motif-level cues that shape community formation. However, most of these methods assume disjoint ground truth and evaluate using partition-based metrics (NMI, ARI), which can understate performance when communities are inherently overlapping.

Renewal Non-Backtracking Random Walks (RNBRW) (Moradi-Jamei et al., 2021) explicitly emphasize cyclic structure by assigning edge reinforcement weights based on retracing statistics from renewal walks that terminate upon forming a cycle. Building on this insight, our proposed *ReCycle Net (RCN)* integrates RNBRW-derived cycle signals into a GAT-style backbone and trains embeddings with a multi-term unsupervised objective, enabling feature-free community detection while more directly leveraging cycle-related structure. Critically, we evaluate RCN on overlapping protein complexes using overlap-aware metrics, demonstrating that appropriate evaluation protocols are necessary to assess complex recovery.

## 3 METHODOLOGY

At its core, RCN builds upon a GAT-based backbone that aggregates features using learned attention mechanisms. However, RCN introduces modifications at both the aggregation and loss stages through RNBRW infusion while replacing node features with one-hot encoded features. This heightens awareness of cyclical structure within communities rather than node features that may not be available in graphs.

RCN avoids the use of explicit supervision or ground-truth labels by operating entirely unsupervised. Community assignments are extracted via k-means clustering on the learned node embeddings, enabling consistent evaluation across datasets of varying scale and label availability. This design allows RCN to perform well on graphs that do not contain node features or labels.

### 3.1 RCN ARCHITECTURE

**Renewal Non-Backtracking Random Walks (RNBRW).** RNBRW assigns edge importance based on recurrence during non-backtracking walks, yielding a weighted adjacency matrix where $w_{ij}$ reflects the frequency with which edge $(i, j)$ participates in cycles Moradi-Jamei et al. (2021). These cycle-aware weights highlight regions of the graph that are structurally significant for community formation.

**Aggregation Function** Standard Graph Attention Networks (GAT) compute attention coefficients purely based on node feature similarity, ignoring higher-order topological signals. To address this, we extend the GAT architecture with a cycle-aware mechanism that integrates RNBRW edge weights directly into the attention computation. RNBRW assigns higher weights to edges that frequently participate in cycles, thereby biasing attention towards edges with higher cyclic structure.

RNBRW is infused in the model at the pre-softmax attention logits with RNBRW-derived weights. It adds a bias term onto the edge before it is softmaxed.

Given node features $h_i \in \mathbb{R}^F$, we compute the standard GAT attention score between nodes $i$ and $j$ as: $e_{ij} = \text{LeakyReLU}(a^\top [W h_i \| W h_j])$, where $W$ is a learnable weight matrix and $a$ is the attention vector. In RCN, we introduce a cycle-aware bias: $\tilde{e}_{ij} = e_{ij} + \log(w_{ij} + \epsilon)$ followed by the usual softmax normalization across the neighborhood $\mathcal{N}(i)$: $\alpha_{ij} = \frac{\exp(\tilde{e}_{ij})}{\sum_{k \in \mathcal{N}(i)} \exp(\tilde{e}_{ik})}$ This formulation allows the network to prioritize messages along edges that are more cycle-relevant, encouraging embeddings that are consistent with the graph's cyclic structure. The final node update rule remains: $h_i' = \sigma \left( \sum_{j \in \mathcal{N}(i)} \alpha_{ij} W h_j \right)$ where $\sigma$ is a non-linear activation function such as ReLU.

**Cycle-Aware Multi-Objective Loss.** To guide the embeddings toward meaningful community structure, RCN is trained with a weighted combination of four unsupervised objectives:

*Cycle-calibrated modularity:* encourages nodes with strong RNBRW-weighted connections to share community labels. $Q \in \mathbb{R}^{n \times C}$ is the soft cluster assignment matrix, $w_{ij}$ is the RNBRW-weighted adjacency, $d_i = \sum_j w_{ij}$, and $m = \frac{1}{2} \sum_{i,j} w_{ij}$: $\mathcal{L}_{\text{mod}} = -\frac{1}{2m} \sum_{i,j} \left( w_{ij} - \frac{d_i d_j}{2m} \right) (Q_i^\top Q_j)$

*Cycle-weighted Laplacian smoothness:* promotes smooth embeddings across RNBRW-weighted edges. $H \in \mathbb{R}^{n \times d}$ is the embedding matrix, $\widetilde{A} = W$ is the RNBRW-weighted adjacency, and $\widetilde{D}$ is its degree matrix: $\mathcal{L}_{\text{lap}} = \text{Tr} \left( H^\top \left( I - \widetilde{D}^{-1/2} \widetilde{A} \widetilde{D}^{-1/2} \right) H \right)$

*Cycle-aligned contrastive learning:* aligns embeddings of top-RNBRW neighbors. $h_i = f(i)$ is the embedding of node $i$, $h_i^+$ is a positive from top RNBRW neighbors, $\text{sim}(\cdot, \cdot)$ is cosine similarity, and $\tau$ is the temperature: $\mathcal{L}_{\text{contrast}} = -\log \left( \frac{\exp(\text{sim}(h_i, h_i^+)/\tau)}{\sum_j \exp(\text{sim}(h_i, h_j)/\tau)} \right)$

*Orthogonality regularization:* prevents head collapse by encouraging near-orthogonal cluster prototypes. $Q^\top Q / n$ approximates the cluster similarity matrix, and $I$ is the identity: $\mathcal{L}_{\text{orth}} = \left\| \frac{Q^\top Q}{n} - I \right\|_F^2$

The total loss combines all components with weights $\lambda_1, \lambda_2, \lambda_3, \lambda_4$: $\mathcal{L} = \lambda_1 \mathcal{L}_{\text{mod}} + \lambda_2 \mathcal{L}_{\text{lap}} + \lambda_3 \mathcal{L}_{\text{contrast}} + \lambda_4 \mathcal{L}_{\text{orth}}$. Each component reinforces a different aspect of community structure, with RNBRW ensuring that all losses are grounded in cycle-aware topology.

**Cluster Selection via $k$-Means**   Community assignments are obtained by applying $k$-means to the learned embeddings. Since ground-truth labels may not align with actual community structure, we estimate the optimal number of clusters $k$ using the elbow method with Silhouette as the primary criterion, and Calinski–Harabasz and Davies–Bouldin indices as tie-breakers. For robustness, we also report results using ground-truth $k$ where available. Full definitions of these validity indices are provided in the Appendix.

### 3.2   Illustrative Mechanism

Figure 1 illustrates a ring of cliques toy graph, where each of the 8 fully connected cliques are linked to its neighbors by a single edge. The RNBRW weights associated with each edge are labeled on the graph. Notably, the bridges between cliques have an RNBRW weight of 0, highlighting that these bridges do not participate in cycles—this structural insight helps RCN overcome the modularity resolution limit. This structure represents a classic resolution-limit case for modularity-based methods: the gain in modularity from keeping small, dense communities separate is too small, leading to their merger. As shown in Figure 1, DMoN collapses multiple cliques into only six coarse groups, failing to preserve the underlying community boundaries. In contrast, RCN correctly identifies all eight cliques due to cycle-aware processing, accurately reflecting the true community structure. Detailed elbow plots and PCA embedding visualizations demonstrating RCN's superior cluster separation compared to DMoN's collapsed structure are provided in Appendix A.1.

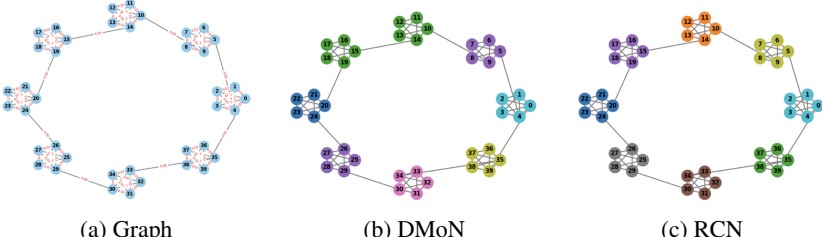

(a) Graph                        (b) DMoN                        (c) RCN

Figure 1: Ring of cliques: RCN identifies all 8 communities vs. DMoN's 6 merged groups.

### 3.3   Cycle-Awareness and Graph Properties

Cycle-awareness is most beneficial when communities exhibit dense *intra*-community cyclic closure (many short cycles) while *inter*-community connections behave more like bridges. In these settings, RNBRW reinforcement weights (Moradi-Jamei et al., 2021) tend to be larger on edges participating in short cycles and smaller on bridge-like edges that rarely close a cycle, which can bias message passing toward structurally cohesive regions.

This suggests that RCN should be most effective on graphs with (i) higher triangle density or clustering coefficient, (ii) abundant short cycles, and (iii) strong cycle–community alignment (cycles concentrate within communities more than across them). Conversely, on graphs where cyclic closure is weaker (lower clustering coefficient and fewer short cycles), cycle-derived signals are expected to provide less additional information beyond standard neighborhood aggregation, and performance gains may be smaller.

From a geometry-grounded perspective, RNBRW yields an edge-weighted graph, which in turn induces a weighted shortest-path metric on vertices. While we do not claim an exact equivalence to a smooth Riemannian metric, this view connects naturally to discrete geometric notions in graphs: cycle closure and local density are related in spirit to curvature-like behavior (e.g., via Ollivier–Ricci curvature (Ollivier, 2009)), and the attention mechanism in GAT-style models (Veličković et al., 2018) can be interpreted as learning anisotropic, structure-aware transport over this weighted geometry.

## 4 EXPERIMENTS

### 4.1 DATASETS & PREPROCESSING.

RCN targets graphs where cycles are important for community formation. Datasets span diverse domains with varying cyclic importance to test RCN's scope:

- **KarateClub (34 nodes):** Friend cycles capture tightly-knit subgroups within the social club.

- **PolBooks (105 nodes):** Co-purchasing cycles among politically similar books reveal ideological clusters.

- **Cora (2708 nodes):** Citation network with sparse cycles—serves as a stress test where cycle-awareness provides limited signal.

- **Facebook Ego (4039 nodes):** Dense friendship triangles and social cycles indicate close-knit communities.

- **Complex Portal PPI (56,944 nodes)**: Cycles correspond to protein complexes or recurring interaction motifs in biological networks. These patterns often signal functional modules, where proteins participate in the same biological processes or pathways.

All datasets were preprocessed by (i) using *feature-free* inputs (ii) symmetrizing edges, and (iii) assigning RNBRW-derived edge weights. RNBRW was run in parallel with a simulation budget sufficient to stabilize the cycle-weight distribution. Once the RNBRW weights were added to the graph, RCN was hyperparameter-tuned to each dataset, and the elbow method was applied to find the optimal k value. To determine the optimal number of clusters, we trained the model with an overcomplete output head and selected the cluster count post hoc from the learned embeddings. For each candidate value of $K$ in the range $[2, K_{\max}]$, we applied $k$-means clustering and evaluated the resulting partitions using Silhouette Rousseeuw (1987), Calinski–Harabasz, Caliński and Harabasz (1974), and Davies–Bouldin Davies and Bouldin (1979). The optimal $K$ was chosen as the value that maximized Silhouette, with Calinski–Harabasz used as a secondary criterion (higher is better) and Davies–Bouldin as a tertiary criterion (lower is better) in cases of ties. This approach can be viewed as a principled extension of the classical Elbow Method: rather than manually inspecting the point at which additional clusters yield diminishing returns, we formalize the choice of $K$ through well-established cluster validity metrics that capture both cohesion and separation. To ensure robustness, we conducted hyperparameter tuning in parallel over a grid of $\lambda_{\mathrm{mod}}, \lambda_{\mathrm{lap}}, \lambda_{\mathrm{contrast}}, \lambda_{\mathrm{orth}}$, and across 3 seeds. Each configuration was trained independently, evaluated with the procedure above, and then aggregated to identify stable hyperparameter settings and consistent estimates of the optimal cluster number.

### 4.2 EXPERIMENTAL SETUP

**Baselines.** We compare RCN against diverse community detection methods: (i) backbone GNNs (GCN, GraphSAGE, GAT), (ii) deep modularity optimization (DMoN), (iii) contrastive approaches (DGI, GRACE, BGRL), (iv) deep clustering models (DAEGC, SDCN), (v) generative overlapping community detection (BigCLAM), and (vi) classical Louvain. We evaluate each method using its native community output when available; otherwise we apply $k$-means to learned embeddings under a common protocol. Additional baseline results (GraphSAGE, DGI, GRACE, BGRL, DAEGC, SDCN, and variants) are reported in Appendix A.4.

**Training.** Models train for 200 epochs using Adam optimizers with learning rates tuned individually per dataset and method. For RCN, both learning rates and loss component weights $\lambda_1, \lambda_2, \lambda_3, \lambda_4$ are optimized via grid search on each dataset to ensure fair comparison. We apply early stopping when validation metrics plateau. Results are averaged across five random seeds for statistical robustness. A complete training procedure is provided in Appendix Algorithm 1.

**Cluster selection.** Primary results use structural $k$ via the elbow–silhouette procedure on embeddings; ties are broken by Calinski–Harabasz (max) and Davies–Bouldin (min). We also report results at ground-truth $k$ where available.

### 4.3 Evaluation Metrics

We evaluate community structure using metrics appropriate to each supervision regime.

**Disjoint ground truth (Karate, PolBooks, Cora).** When disjoint community labels are available, we report *NMI* and *ARI* as the primary external metrics of agreement between predicted partitions and ground truth. We additionally report *Silhouette* as an internal measure of embedding separation.

**Unlabeled graphs (Facebook Ego).** Since no ground-truth labels are available, we report internal clustering validity index *Silhouette* as the primary quantitative criteria for selecting and evaluating the number of communities.

**Overlapping ground truth (PPI, Complex Portal).** Because proteins may belong to multiple complexes, standard partition metrics (NMI/ARI/Silhouette) are not appropriate as primary measures. We therefore use *Overlapping NMI (ONMI)* and evaluate predicted covers constructed via a top-$r$ membership rule, masking unlabeled nodes and excluding singleton communities (size $< 2$) in both ground truth and predictions.

Finally, we note that benchmark labels may not always coincide with the most structurally coherent partition of a graph. For this reason, internal indices (Silhouette/CH/DB) are used for model selection when labels are missing and are reported as complementary diagnostics when labels exist.

**Computational cost.** RCN adds a one-time RNBRW preprocessing step to compute cycle-informed edge weights. RNBRW is a scalable random-walk-based procedure whose cost is linear in the number of simulated renewal walks and can be parallelized across CPU cores (Moradi-Jamei et al., 2021). After weights are computed, training cost is comparable to a standard GAT-style encoder because RNBRW enters only as fixed edge weights in message passing; thus, the dominant runtime in our pipeline is GNN training, while RNBRW preprocessing is amortized across multiple runs on the same graph.

**Computational complexity.** RCN adds a one-time RNBRW preprocessing step to estimate cycle-informed edge weights $w_{ij}$. RNBRW is a random-walk simulation whose runtime is linear in the total number of simulated steps; with $\rho$ renewal walks and expected walk length $\ell$, the expected preprocessing cost is $O(\rho\ell)$ and memory is $O(|E|)$ to store edge weights. After preprocessing, training has the same order as a standard multi-head, $L$-layer GAT: per epoch the dominant cost is sparse attention/message passing over edges, $O(L\,H\,|E|\,d)$ up to constants, since RNBRW enters only as fixed edge weights/bias terms in attention. With the contrastive objective, each epoch typically requires $V$ augmented forward passes (we use V=2V=2V=2), so the dominant message-passing cost becomes $O(V\,L\,H\,|E|\,d)$, plus the contrastive similarity computation (implementation-dependent; typically linear in batch size and embedding dimension). Community extraction by $k$-means adds $O(I\,n\,K\,d)$ per candidate $K$ (with $I$ Lloyd iterations), and validity-based $K$ selection scales linearly with the number of $K$ values evaluated. A concise comparison to common baselines is provided in Appendix B.1.

**Hyperparameterization and robustness.** RCN uses a weighted multi-objective loss with four coefficients. To limit tuning, we search over a small discrete grid per dataset and report robustness via a one-factor-at-a-time sensitivity analysis on our flagship PPI setting (Appendix B.5). This shows performance is stable under moderate rescaling of most terms, with the largest sensitivity arising from the contrastive coefficient.

## 5 Results

We evaluate RCN on Karate Club, PolBooks, Cora, and Facebook, and a PPI benchmark from Complex Portal. Results are reported for both (i) $k$ estimated via the elbow–silhouette procedure (Table 1) and (ii) $k$ set to the number of ground-truth communities (Table 1). All values are averaged across five runs.

**Ground Truth vs. Structural $k$** A key observation is that silhouette scores are consistently higher when $k$ is chosen by a structural criterion (elbow–silhouette) rather than forced to match ground truth. For example, on *Karate Club*, RCN attains a silhouette of $0.81\pm0.01$ under structural $k$; forcing $k{=}2$

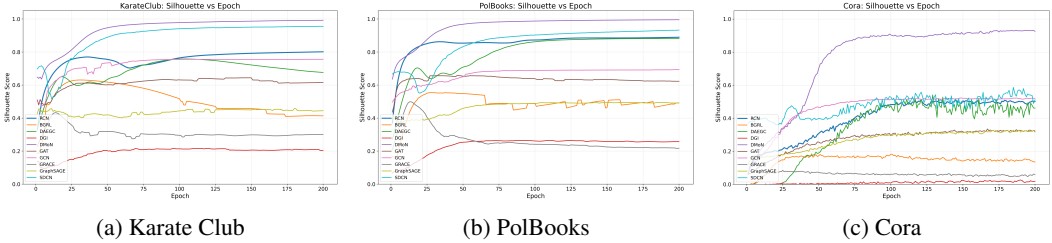

(a) Karate Club                        (b) PolBooks                        (c) Cora

Figure 2: Silhouette score convergence across datasets (higher is better).

Table 1: Community detection results (mean $\pm$ std over 5 runs). The left block uses a structural choice of $k$ (elbow–silhouette), while the right block uses ground-truth $k$ when available. Higher is better. For the full suite of baselines and ablations, see Appendix A.4.

| Dataset | Model | Structural $k$ | | | Ground-truth $k$ | | |
|---|---|---|---|---|---|---|---|
| | | NMI | ARI | Sil. | NMI | ARI | Sil. |
| Karate | GAT | $0.22 \pm 0.16$ | $0.20 \pm 0.23$ | $0.62 \pm 0.17$ | $0.22 \pm 0.16$ | $0.20 \pm 0.23$ | $0.62 \pm 0.17$ |
| | GCN | $0.01 \pm 0.00$ | $-0.01 \pm 0.00$ | $0.76 \pm 0.01$ | $0.01 \pm 0.00$ | $-0.01 \pm 0.00$ | $0.76 \pm 0.01$ |
| | Louvain | $0.65 \pm 0.01$ | $0.66 \pm 0.02$ | $-$ | $0.65 \pm 0.01$ | $0.66 \pm 0.02$ | $-$ |
| | DMoN | $0.84 \pm 0.00$ | $0.88 \pm 0.00$ | $0.99 \pm 0.00$ | $0.84 \pm 0.00$ | $0.88 \pm 0.00$ | $0.99 \pm 0.00$ |
| | RCN | $0.84 \pm 0.00$ | $0.88 \pm 0.00$ | $0.81 \pm 0.01$ | $0.84 \pm 0.00$ | $0.88 \pm 0.00$ | $0.81 \pm 0.01$ |
| PolBooks | GAT | $0.16 \pm 0.13$ | $0.18 \pm 0.16$ | $0.62 \pm 0.12$ | $0.39 \pm 0.19$ | $0.41 \pm 0.25$ | $0.54 \pm 0.10$ |
| | GCN | $0.01 \pm 0.01$ | $-0.02 \pm 0.01$ | $0.69 \pm 0.01$ | $0.01 \pm 0.01$ | $-0.02 \pm 0.00$ | $0.54 \pm 0.01$ |
| | Louvain | $0.58 \pm 0.00$ | $0.62 \pm 0.00$ | $-$ | $-$ | $-$ | $-$ |
| | DMoN | $0.60 \pm 0.00$ | $0.67 \pm 0.00$ | $0.99 \pm 0.00$ | $0.58 \pm 0.01$ | $0.67 \pm 0.01$ | $0.97 \pm 0.00$ |
| | RCN | $0.60 \pm 0.00$ | $0.67 \pm 0.00$ | $0.89 \pm 0.00$ | $0.55 \pm 0.02$ | $0.62 \pm 0.02$ | $0.77 \pm 0.03$ |
| Cora | GAT | $0.04 \pm 0.00$ | $0.01 \pm 0.00$ | $0.32 \pm 0.05$ | $0.03 \pm 0.01$ | $0.01 \pm 0.00$ | $0.44 \pm 0.02$ |
| | GCN | $0.01 \pm 0.00$ | $0.01 \pm 0.00$ | $0.47 \pm 0.52$ | $0.01 \pm 0.00$ | $0.01 \pm 0.00$ | $0.52 \pm 0.01$ |
| | Louvain | $0.36 \pm 0.01$ | $0.23 \pm 0.01$ | $-$ | $-$ | $-$ | $-$ |
| | DMoN | $0.22 \pm 0.05$ | $0.15 \pm 0.05$ | $0.93 \pm 0.02$ | $0.17 \pm 0.05$ | $0.11 \pm 0.05$ | $0.93 \pm 0.02$ |
| | RCN | $0.38 \pm 0.01$ | $0.27 \pm 0.02$ | $0.50 \pm 0.02$ | $0.36 \pm 0.03$ | $0.26 \pm 0.04$ | $0.38 \pm 0.04$ |

(ground truth) reduces silhouette despite similar ARI/NMI (Table 1). This indicates that benchmark label taxonomies need not coincide with the most structurally coherent partition, underscoring the value of structural validity indices—Silhouette Rousseeuw (1987), Calinski–Harabasz Caliński and Harabasz (1974), and Davies–Bouldin Davies and Bouldin (1979)—alongside label-based metrics. In practice, we choose $k$ via elbow–silhouette on the learned (cycle-aware) embeddings and treat ARI/NMI as reference measures when labels are class-oriented rather than community-grounded.

**Cycle-Rich Graphs: Karate Club and PolBooks**    On cycle-rich graphs, RCN performs strongly relative to baselines. On *Karate Club*, RCN ties with DMoN for the best ARI ($0.88$) and NMI ($0.84$) and attains a silhouette of $0.81$, outperforming backbone GNNs such as GAT (ARI $0.20$, NMI $0.22$) and GCN (ARI $0.39$, NMI $0.39$) (Table 1). On *PolBooks*, RCN achieves ARI $0.67$ and NMI $0.60$ with a high silhouette of $0.89$, exceeding most baselines under structural $k$. DMoN attains near-perfect silhouette on Karate and PolBooks, while RCN matches or is competitive on NMI/ARI; we therefore treat silhouette as an internal geometry diagnostic rather than a sole indicator of label agreement on labeled benchmarks. These results support the hypothesis that RNBRW-based cycle reinforcement provides a strong signal for community detection when short cycles are prevalent.

**Tree-Like Graphs: Cora**    *Cora* serves as a stress test for our scope: there, cycles are not particularly important in forming communities, so the RNBRW bias carries limited signal. Under structural $k$ (Table 1), RCN attains NMI $0.38$, ARI $0.27$, and silhouette $0.50$, competitive with GCN/GAT and slightly below DMoN (NMI $0.39$, ARI $0.28$). When the ground-truth $k=7$ is enforced (Table 1), RCN's ARI/NMI shift to $0.26/0.36$ while silhouette drops to $0.38$, echoing the labels–structure mismatch discussed in Sec. 4.1. Figure 2 shows stable convergence for RCN but smaller silhouette gains than on cycle-rich graphs (*Karate Club*, *PolBooks*). We also tested RCN using Cora's features. It did not improve over feature-free (Appendix B.4).

**Large-Scale Graph: Facebook Ego Network**    Because Facebook lacks ground-truth labels, we select the number of communities using structural criteria. On the ego network ($\sim$4k nodes, 88k

Table 2: Complex Portal PPI: overlap-aware evaluation using McDaid–Greene–Hurley ONMI. We report partition ONMI ($r = 1$) and the best overlap ONMI over $r \in \{1, \ldots, 5\}$ (top-$r$ nearest centroids). Results are mean $\pm$ std. *CB refers to "Cycle-Broken". For BigCLAM, we evaluate the native overlapping cover (no top-r conversion)

| Method | $K$ | ONMI ($r = 1$) | Best ONMI | $r^{\star}$ | AvgMem | #PredComms$_{\geq 2}$ |
|--------|-----|----------------|-----------|-------------|--------|----------------------|
| DMoN | 793 | 0.108±0.005 | 0.140±0.003 | 5 | 5.0 | 176±3 |
| GAT | 793 | 0.175±0.010 | 0.232±0.003 | 2 | 2.0 | 356±4 |
| BigCLAM | 793 | - | 0.243±0.000 | - | 10.0 | 789±0 |
| RCN (CB*) | 793 | 0.264±0.003 | 0.317±0.0017 | 2 | 2.0 | 514±3 |
| RCN | 793 | 0.313±0.002 | 0.344±0.005 | 2 | 2.0 | 560±2 |

edges), the elbow–silhouette procedure identifies $k=5$ for RCN (silhouette $= 0.85$), and the t-SNE embedding (Fig. 3) exhibits compact, well-separated clusters with visible substructure consistent with cycle-preserving organization. In contrast, DMoN yields conflicting diagnostics—the elbow suggests $k=2$ while the silhouette peaks at $k=4$ (0.99)—and its embeddings collapse into elongated, filamentary manifolds, exaggerating separation while losing fine cyclic detail. Overall, RCN preserves both macro-level partitions and micro-level cycle-aware sub-communities, whereas DMoN favors fewer clusters with distorted geometry. Full elbow and silhouette plots for both methods are provided in Appendix A.3 as well as all baseline performances in Appendix A.4.

**PPI (Complex Portal, overlapping complexes).** Protein complexes form densely interdependent interaction patterns where short cycles are common. Since Complex Portal ground truth is overlapping, single-membership partitions can understate recovery quality. We therefore evaluate RCN using overlap-aware ONMI under a top-$r$ membership rule. On the Complex Portal PPI graph, partition evaluation ($r = 1$) increases up to $K = 793$, where ONMI reaches $0.313 \pm 0.002$. Allowing mild overlap improves agreement with the multi-membership nature of complexes: the best setting is $K = 793$ with $r = 2$, achieving ONMI $= 0.344 \pm 0.005$ with AvgMem $= 2.0$. A post hoc $K$ sweep (re-running $k$-means on fixed embeddings) further shows a granularity–stability tradeoff: larger $K$ produces many small predicted communities (at $K = 793$, $560 \pm 2$ predicted communities survive a size $\geq 2$ filter, with median size 3 and 90th percentile size 9), while overlap-aware evaluation remains more robust than forced partitions. Compared to baselines, RCN substantially improves overlap recovery under the same evaluation protocol: at $K = 793$, the best-over-$r \in \{1, \ldots, 5\}$ ONMI is $0.344 \pm 0.005$ (at $r = 2$), versus $0.232 \pm 0.003$ for a standard GAT, $0.140 \pm 0.003$ for DMoN, and $0.243\pm0.000$ for BigCLAM. These results support the view that cycle-informed embeddings are useful on PPI graphs, and that overlap-aware evaluation is necessary to fairly assess complex recovery.

**Mechanism control: cycle breaking.** To isolate whether RNBRW is leveraging cyclic structure rather than generic degree effects, we construct a degree-preserving *cycle-broken* surrogate of the Complex Portal PPI graph via random edge swaps, recompute RNBRW weights on the surrogate, and transfer the resulting weights back to the original edge set for training and evaluation. At fixed $K = 793$ and overlap evaluation $r = 2$, full RNBRW achieves ONMI $= 0.344 \pm 0.005$ (two seeds), whereas cycle-broken RNBRW weights reduce performance to ONMI $= 0.317 \pm 0.0017$, a drop of $\Delta$ONMI $= 0.0292 \pm 0.0018$. This degradation under cycle destruction supports the claim that RNBRW contributes signal tied to short-cycle structure in PPI complexes rather than acting as a purely degree-based reweighting.

**Lambda sensitivity (PPI).** We performed a one-factor-at-a-time sensitivity analysis on Complex Portal PPI using our best configuration, fixing evaluation to $K = 793$ and overlap scoring at $r = 2$ (two seeds: 120, 42). Scaling each loss weight by $\{0, 0.5, 1, 2\}$ while holding the others fixed shows that performance is broadly stable for moderate rescalings: ONMI remains in the $\approx 0.34$ range for multipliers in $\{0.5, 1, 2\}$ for $\lambda_{mod}$, $\lambda_{lap}$, and $\lambda_{orth}$. In contrast, removing the contrastive term ($\lambda_{con} = 0$) causes a drop to ONMI $= 0.331 \pm 0.007$ and produces substantially more tiny predicted communities (fewer survive the size $\geq 2$ filter), indicating that contrastive regularization is critical for non-degenerate cycle-aware embeddings on PPI. Full sensitivity results are reported in Appendix B.5.

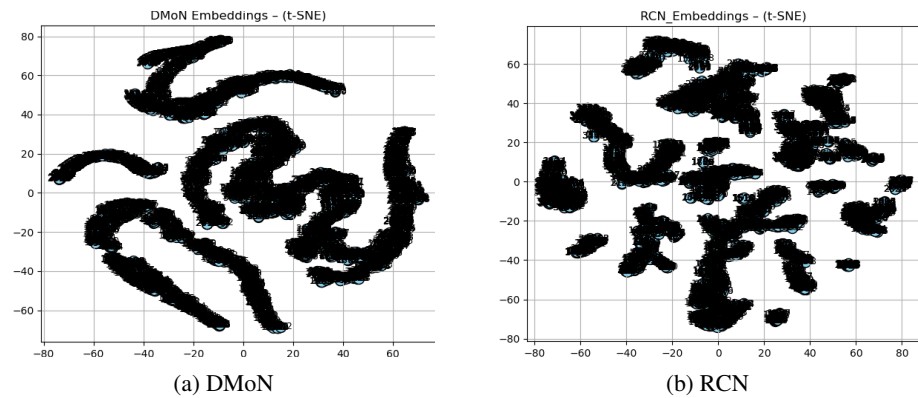

Figure 3: Facebook ego network (unlabeled): RCN vs. DMoN embeddings

## 6 CONCLUSION

We introduce ReCycle Net (RCN), a feature-free, cycle-aware GNN that injects RNBRW into GAT attention and trains under a compact multi-objective loss. On graphs where cycles inform community formation, RCN performs competitively with strong baselines and yields coherent structure (Karate, PolBooks; Facebook: structural $k=5$, silhouette $0.85$), while remaining competitive on cycle-sparse Cora—showing that explicitly modeling cyclic motifs improves community coherence and interpretability without attributes or labels.

**Limitations:** RNBRW preprocessing adds overhead; gains are largest when cycles drive communities; multi-objective weights require tuning.

**Future work:** streaming/sampled RNBRW with mini-batch training; GraphSAGE-style neighborhood sampling (*RCN-SAGE*); PPI evaluation where cycles are impactful (ONMI, enrichment); overlapping memberships; dynamic/heterogeneous graphs; ablations of loss terms and RNBRW bias.

**Reproducibility:** Code, RNBRW scripts, and plotting utilities with per-seed results will be released upon publication.

**Broader impacts:** Potential benefits include biological and biomedical graphs where cycles mark functional modules (e.g., protein complexes/feedback loops); on social data, mitigate profiling risks via consent, anonymization, and audits.

*Acknowledgments*—This material is based upon work supported by the National Science Foundation under Grant No. 2349593. Any opinions, findings, and conclusions or recommendations expressed in this material are those of the author(s) and do not necessarily reflect the views of the National Science Foundation. The initial phase of this work was carried out during the REU program supported by this grant.

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

# A  ADDITIONAL EXPERIMENTAL ANALYSIS

## A.1  TOY GRAPH ANALYSIS: DETAILED RESULTS

Figure 4 provides comprehensive analysis of the ring-of-cliques toy graph, demonstrating RCN's ability to preserve cycle-aware structure while avoiding the over-collapsing behavior exhibited by DMoN.

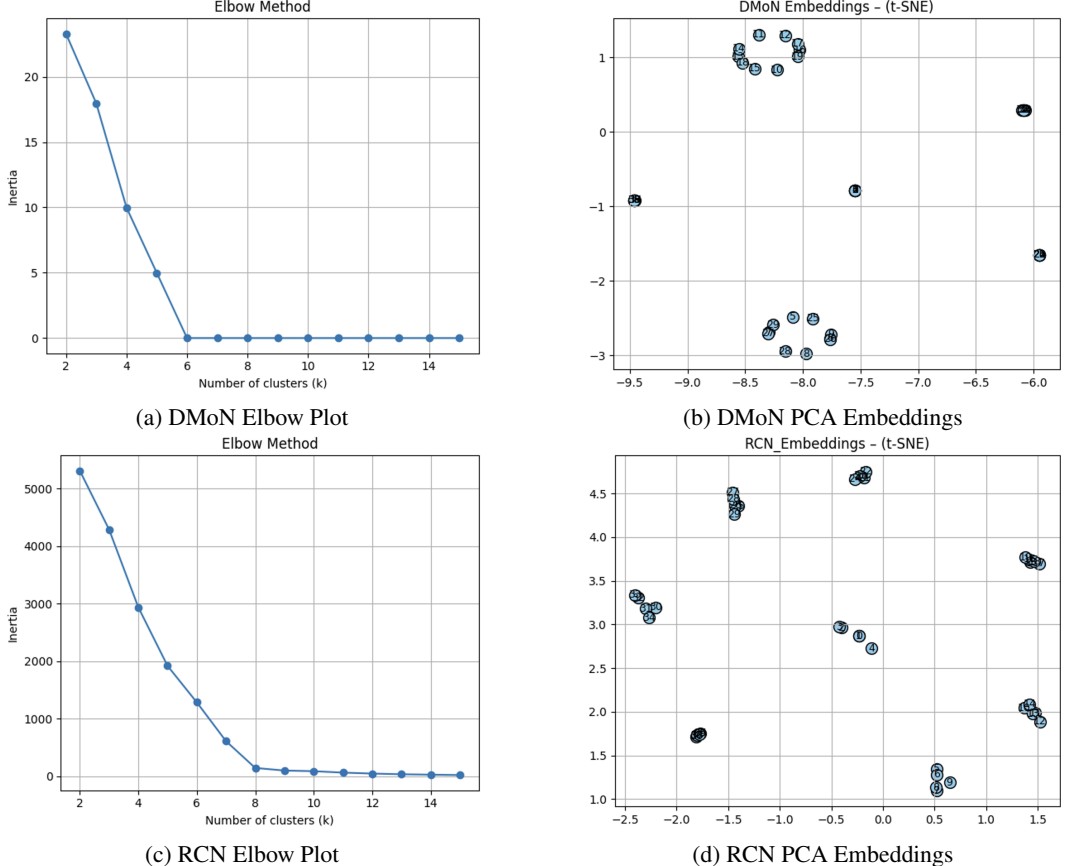

Figure 4: Detailed toy graph analysis showing RCN's superior cluster separation compared to DMoN's collapsed structure.

## A.2  TRAINING DYNAMICS AND CONVERGENCE

Figure 5 demonstrates stable optimization behavior across all datasets, with particularly effective convergence on cycle-rich graphs where the multi-objective loss successfully captures both cycle-aware and community structure signals.

## A.3  FACEBOOK EGO NETWORK: DETAILED ANALYSIS

Figure 6 provides detailed cluster selection diagnostics for the Facebook ego network, illustrating the contrast between RCN's coherent structural organization and DMoN's conflicting metrics.

## A.4  ADDITIONAL BASELINES AND ABLATIONS

### A.4.1  HYPERPARAMETER SEARCH DETAILS

**Karate.** $\lambda_{\text{mod}} = \{0.3, 0.6, 0.9\}$, $\lambda_{\text{con}}, \lambda_{\text{lap}} \in \{10^{-3}, 10^{-4}, 10^{-5}\}$, $\lambda_{\text{orth}} \in \{10^{-3}, 10^{-4}, 10^{-5}, 10^{-6}\}$

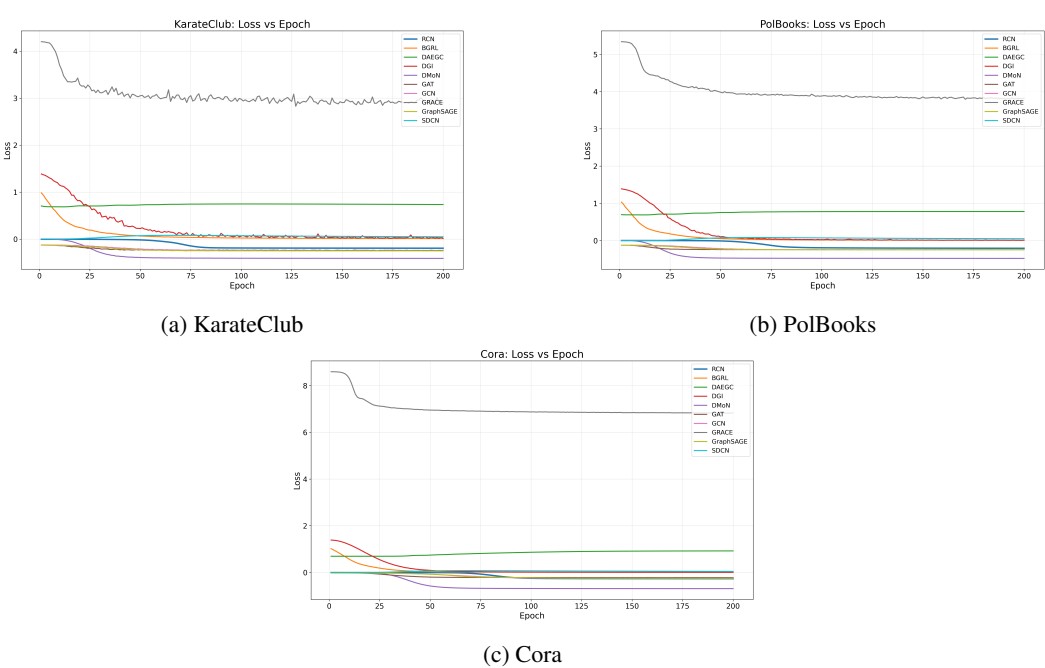

Figure 5: Loss convergence curves showing stable optimization across datasets.

**PolBooks.** $\lambda_{\text{mod}} = \{0.3, 0.6, 0.9\}$, $\lambda_{\text{con}}, \lambda_{\text{lap}} \in \{10^{-3}, 10^{-4}, 10^{-5}\}$, $\lambda_{\text{orth}} \in \{10^{-3}, 10^{-4}, 10^{-5}, 10^{-6}\}$

**Cora.** $\lambda_{mod} = \{0.3, 0.6, 0.9\}$, $\lambda_{con}, \lambda_{lap} \in \{10^{-3}, 10^{-4}, 10^{-5}\}$, $\lambda_{orth} \in \{10^{-3}, 10^{-4}, 10^{-5}, 10^{-6}\}$

**FacebookEgo.** $\lambda_{mod} \in \{0.3, 0.6, 0.9\}$, $\lambda_{con}, \lambda_{lap} \in \{10^{-3}, 10^{-4}, 10^{-5}\}$, $\lambda_{orth} \in \{10^{-3}, 10^{-4}, 10^{-5}, 10^{-6}\}$

**PPI.** $\lambda_{mod} \in \{0.05, 0.1, 0.2\}$, $\lambda_{con} \in \{0.1, 5 \times 10^{-2}, 10^{-3}, 5 \times 10^{-3}\}$, $\lambda_{lap} \in \{10^{-5}, 10^{-6}, 10^{-7}\}$, $\lambda_{orth} \in \{10^{-5}, 10^{-6}\}$

# B IMPLEMENTATION DETAILS

## B.1 COMPUTATIONAL COMPLEXITY SUMMARY

## B.2 REPRESENTATIVE RESOURCE TABLE

## B.3 RCN TRAINING PROCEDURE

## B.4 CORA FEATURE VS. ONE-HOT ABLATION

## B.5 LAMBDA SENSITIVITY ON PPI

## B.6 PPI CLUSTERING COEFFICIENT AND TRIANGLE COUNTS

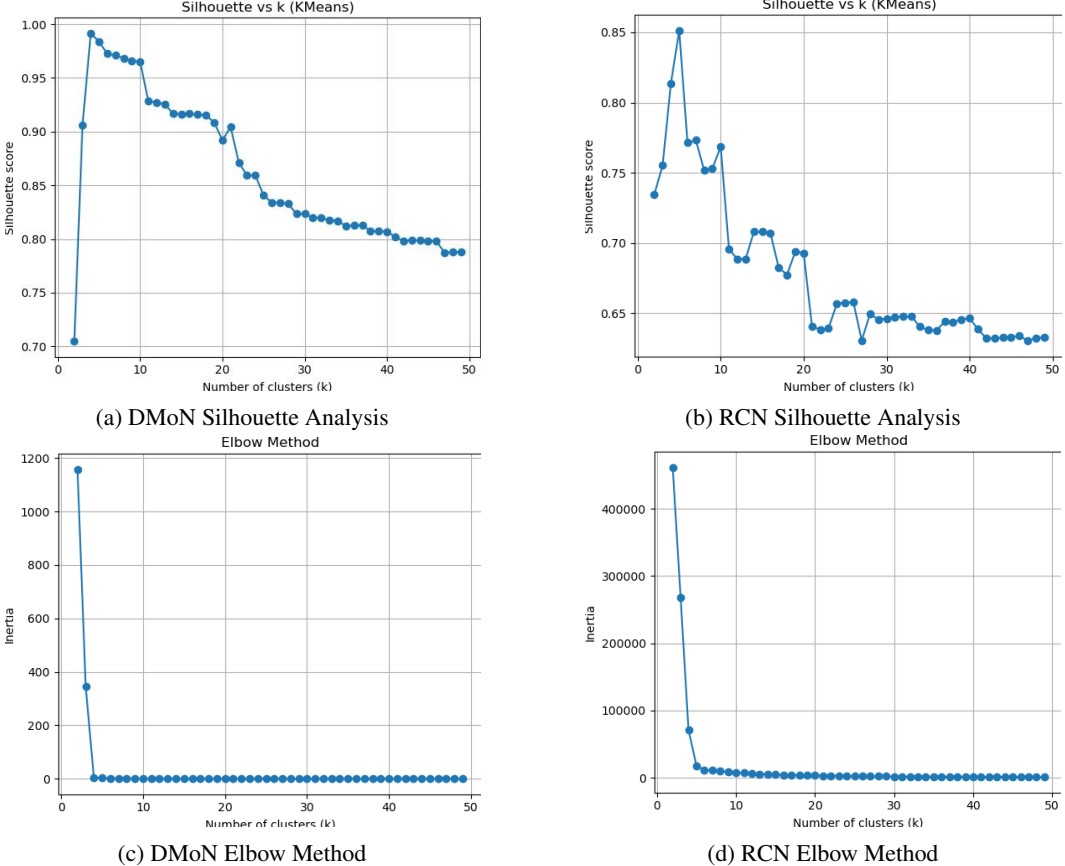

(a) DMoN Silhouette Analysis

(b) RCN Silhouette Analysis

(c) DMoN Elbow Method

(d) RCN Elbow Method

Figure 6: Facebook ego network cluster selection analysis revealing RCN's consistent diagnostics versus DMoN's conflicting signals.

| Dataset | Model | Structural $k$ | | | Ground-truth $k$ | | |
|---|---|---|---|---|---|---|---|
| | | NMI | ARI | Sil. | NMI | ARI | Sil. |
| Karate Club | BGRL | $0.72 \pm 0.09$ | $0.75 \pm 0.11$ | $0.41 \pm 0.04$ | $0.72 \pm 0.09$ | $0.75 \pm 0.11$ | $0.41 \pm 0.04$ |
| | DAEGC | $0.82 \pm 0.05$ | $0.86 \pm 0.05$ | $0.68 \pm 0.01$ | $0.82 \pm 0.05$ | $0.86 \pm 0.05$ | $0.68 \pm 0.01$ |
| | DGI | $0.68 \pm 0.16$ | $0.71 \pm 0.22$ | $0.20 \pm 0.02$ | $0.68 \pm 0.16$ | $0.71 \pm 0.22$ | $0.20 \pm 0.02$ |
| | DMoN | $0.84 \pm 0.00$ | $0.88 \pm 0.00$ | $0.99 \pm 0.00$ | $0.84 \pm 0.00$ | $0.88 \pm 0.00$ | $0.99 \pm 0.00$ |
| | GAT | $0.22 \pm 0.16$ | $0.20 \pm 0.23$ | $0.62 \pm 0.17$ | $0.22 \pm 0.16$ | $0.20 \pm 0.23$ | $0.62 \pm 0.17$ |
| | GCN | $0.01 \pm 0.00$ | $-0.01 \pm 0.00$ | $0.76 \pm 0.01$ | $0.01 \pm 0.00$ | $-0.01 \pm 0.00$ | $0.76 \pm 0.01$ |
| | GRACE | $0.82 \pm 0.05$ | $0.86 \pm 0.05$ | $0.30 \pm 0.01$ | $0.82 \pm 0.05$ | $0.86 \pm 0.05$ | $0.30 \pm 0.01$ |
| | GraphSAGE | $0.14 \pm 0.16$ | $0.15 \pm 0.19$ | $0.44 \pm 0.02$ | $0.14 \pm 0.16$ | $0.15 \pm 0.19$ | $0.44 \pm 0.02$ |
| | Louvain (Unweighted) | $0.50 \pm 0.00$ | $0.43 \pm 0.00$ | $-$ | $0.50 \pm 0.00$ | $0.43 \pm 0.00$ | $-$ |
| | Louvain (Weighted) | $0.59 \pm 0.00$ | $0.52 \pm 0.00$ | $-$ | $0.59 \pm 0.00$ | $0.52 \pm 0.00$ | $-$ |
| | RCN | $0.84 \pm 0.00$ | $0.88 \pm 0.00$ | $0.81 \pm 0.01$ | $0.84 \pm 0.00$ | $0.88 \pm 0.00$ | $0.81 \pm 0.01$ |
| | SDCN | $0.21 \pm 0.27$ | $0.22 \pm 0.32$ | $0.95 \pm 0.03$ | $0.21 \pm 0.27$ | $0.22 \pm 0.32$ | $0.95 \pm 0.03$ |
| PolBooks | BGRL | $0.61 \pm 0.01$ | $0.67 \pm 0.01$ | $0.49 \pm 0.13$ | $0.53 \pm 0.05$ | $0.57 \pm 0.07$ | $0.50 \pm 0.09$ |
| | DAEGC | $0.59 \pm 0.03$ | $0.66 \pm 0.03$ | $0.88 \pm 0.01$ | $0.50 \pm 0.01$ | $0.54 \pm 0.03$ | $0.80 \pm 0.10$ |
| | DGI | $0.54 \pm 0.06$ | $0.59 \pm 0.08$ | $0.26 \pm 0.02$ | $0.54 \pm 0.04$ | $0.61 \pm 0.07$ | $0.26 \pm 0.03$ |
| | DMoN | $0.60 \pm 0.00$ | $0.67 \pm 0.00$ | $1.00 \pm 0.00$ | $0.58 \pm 0.01$ | $0.67 \pm 0.01$ | $0.97 \pm 0.00$ |
| | GAT | $0.16 \pm 0.13$ | $0.18 \pm 0.16$ | $0.62 \pm 0.12$ | $0.39 \pm 0.19$ | $0.41 \pm 0.25$ | $0.54 \pm 0.10$ |
| | GCN | $0.01 \pm 0.01$ | $-0.02 \pm 0.01$ | $0.69 \pm 0.01$ | $0.01 \pm 0.01$ | $-0.02 \pm 0.00$ | $0.54 \pm 0.01$ |
| | GRACE | $0.47 \pm 0.14$ | $0.51 \pm 0.14$ | $0.22 \pm 0.01$ | $0.43 \pm 0.14$ | $0.46 \pm 0.17$ | $0.22 \pm 0.05$ |
| | GraphSAGE | $0.04 \pm 0.03$ | $0.03 \pm 0.05$ | $0.49 \pm 0.03$ | $0.05 \pm 0.02$ | $0.03 \pm 0.02$ | $0.42 \pm 0.03$ |
| | Louvain (Unweighted) | $0.57 \pm 0.00$ | $0.68 \pm 0.00$ | $-$ | $0.57 \pm 0.00$ | $0.68 \pm 0.00$ | $-$ |
| | Louvain (Weighted) | $0.53 \pm 0.00$ | $0.63 \pm 0.00$ | $-$ | $0.50 \pm 0.00$ | $0.56 \pm 0.00$ | $-$ |
| | RCN | $0.60 \pm 0.00$ | $0.67 \pm 0.00$ | $0.89 \pm 0.00$ | $0.55 \pm 0.02$ | $0.62 \pm 0.02$ | $0.77 \pm 0.03$ |
| | SDCN | $0.03 \pm 0.02$ | $0.02 \pm 0.03$ | $0.93 \pm 0.01$ | $0.03 \pm 0.01$ | $0.01 \pm 0.02$ | $0.78 \pm 0.04$ |
| Cora | BGRL | $0.01 \pm 0.01$ | $0.00 \pm 0.00$ | $0.14 \pm 0.03$ | $0.01 \pm 0.00$ | $0.01 \pm 0.00$ | $0.22 \pm 0.06$ |
| | DAEGC | $0.08 \pm 0.04$ | $0.05 \pm 0.03$ | $0.46 \pm 0.21$ | $0.06 \pm 0.03$ | $0.04 \pm 0.02$ | $0.53 \pm 0.23$ |
| | DGI | $0.08 \pm 0.02$ | $0.04 \pm 0.01$ | $0.02 \pm 0.01$ | $0.06 \pm 0.01$ | $0.03 \pm 0.01$ | $0.01 \pm 0.01$ |
| | DMoN | $0.22 \pm 0.05$ | $0.15 \pm 0.05$ | $0.93 \pm 0.02$ | $0.17 \pm 0.05$ | $0.11 \pm 0.05$ | $0.93 \pm 0.02$ |
| | GAT | $0.04 \pm 0.00$ | $0.01 \pm 0.00$ | $0.32 \pm 0.05$ | $0.03 \pm 0.01$ | $0.01 \pm 0.00$ | $0.44 \pm 0.02$ |
| | GCN | $0.01 \pm 0.00$ | $0.01 \pm 0.00$ | $0.52 \pm 0.01$ | $0.01 \pm 0.00$ | $0.01 \pm 0.00$ | $0.52 \pm 0.01$ |
| | GRACE | $0.26 \pm 0.04$ | $0.16 \pm 0.04$ | $0.06 \pm 0.01$ | $0.17 \pm 0.05$ | $0.10 \pm 0.04$ | $0.04 \pm 0.01$ |
| | GraphSAGE | $0.03 \pm 0.01$ | $0.00 \pm 0.00$ | $0.32 \pm 0.04$ | $0.02 \pm 0.00$ | $0.00 \pm 0.00$ | $0.40 \pm 0.03$ |
| | Louvain (Unweighted) | $0.46 \pm 0.00$ | $0.26 \pm 0.00$ | $-$ | $0.46 \pm 0.00$ | $0.26 \pm 0.00$ | $-$ |
| | Louvain (Weighted) | $0.42 \pm 0.00$ | $0.13 \pm 0.00$ | $-$ | $0.42 \pm 0.00$ | $0.13 \pm 0.00$ | $-$ |
| | RCN | $0.38 \pm 0.01$ | $0.27 \pm 0.02$ | $0.50 \pm 0.02$ | $0.36 \pm 0.03$ | $0.26 \pm 0.04$ | $0.38 \pm 0.04$ |
| | SDCN | $0.01 \pm 0.00$ | $0.01 \pm 0.00$ | $0.51 \pm 0.09$ | $0.01 \pm 0.00$ | $0.00 \pm 0.00$ | $0.61 \pm 0.14$ |

Table 3: Additional baseline results (mean $\pm$ std). Structural $k$ uses elbow–silhouette; ground-truth $k$ uses labeled $k$ when available.

| Method | Silhouette |
|---|---|
| GRACE | $0.1545 \pm 0.0256$ |
| DGI | $0.1715 \pm 0.0132$ |
| GraphSAGE | $0.3782 \pm 0.0483$ |
| GAT | $0.3950 \pm 0.0846$ |
| BGRL | $0.4038 \pm 0.0497$ |
| GCN | $0.5235 \pm 0.0186$ |
| SDCN | $0.6479 \pm 0.0960$ |
| DAEGC | $0.7460 \pm 0.1939$ |
| RCN | $0.8742 \pm 0.0388$ |
| DMoN | $0.9610 \pm 0.0134$ |

Table 4: Silhouette performance of baseline models on Facebook Ego. RCN performs competitively and further analysis as seen in 3 compares RCN embeddings against DMoN (the top-performer).

Table 5: Hyperparameter tuning budget per dataset. Configurations denotes the number of hyperparameter combinations evaluated, Seeds indicates the random initializations used per configuration, and Total Runs equals configurations multiplied by seeds. The final configuration was selected using the metrics listed in the Selection Metric column.

| Dataset | Seeds | Configurations | Total Runs | Selection Metric |
|---|---|---|---|---|
| Karate Club | 120, 42, 328476 | 108 | 324 | ARI, NMI, Sil |
| PolBooks | 120, 42, 328476 | 108 | 324 | ARI, NMI, Sil |
| Cora | 120, 42, 328476 | 108 | 324 | ARI, NMI, Sil |
| FacebookEgo | 120, 42, 328476 | 108 | 324 | Sil |
| PPI | 120, 42 | 72 | 144 | ONMI, Sil |

| Method | One-time preprocessing | Per-epoch / main runtime |
|---|---|---|
| RCN | RNBRW: $O(\rho\ell)$, mem $O(|E|)$ | GAT-like: $O(V\,L\,H\,|E|\,d) + \mathcal{C}_{\text{con}}$ |
| GAT | none | $O(L\,H\,|E|\,d)$ |
| DMoN | none | GNN encoder $+ O(nK)$ assignment head (impl.-dependent) |
| Louvain | none | $\tilde{O}(|E|)$ per pass (heuristic) |
| BigCLAM | none | iterative; depends on membership sparsity |

Table 6: High-level computational cost comparison. Here $n = |V|$, $|E|$ is number of edges, $d$ hidden width, $L$ layers, $H$ attention heads, $\rho$ renewal walks, and $\ell$ expected walk length. $V$ is the number of augmented views used by the contrastive objective (typically $V = 2$). $\mathcal{C}_{\text{con}}$ denotes the additional contrastive similarity computation, which is implementation-dependent (often linear in the number of sampled pairs and $d$).

Table 7: Resource usage comparison: RCN (HPC) vs BigCLAM (local) on PPI dataset. This is representative. RCN was run on the University of Virginia's Rivanna HPC while BigCLAM was run on a AMD Threadripper 7970x local workstation.

| Method | Elapsed | MaxRSS | AllocCPUS |
|---|---|---|---|
| RCN (HPC, Python) | 07:17:41 | 19.52G | 1 |
| BigCLAM (Local, C++) | 0:10:04.78 | 46344 kB (0.0442 GB) | 32 |

---

**Algorithm 1:** Algorithm 1: ReCycle Net (RCN) training procedure

---

**Input:** Graph $G = (V, E)$; node features $X \in \mathbb{R}^{N \times F}$; RNBRW retrace probabilities
$\quad$ $w_{ij} \in [0, 1]$ for $(i, j) \in E$; number of communities $K$; loss weights $\lambda_{\text{mod}}, \lambda_{\text{lap}}, \lambda_{\text{con}}, \lambda_{\text{orth}}$;
$\quad$ temperatures $T$ (attention) and $\tau$ (contrastive)

**Output:** Hard community labels $\hat{y} \in \{1, \ldots, K\}^N$

**Preprocessing:** Duplicate each undirected edge $(i, j) \in E$ into $(i \to j)$ and $(j \to i)$

**Cycle-aware attention (per layer): foreach** *directed edge $(i \to j)$ and head $h$* **do**

$$e_{ij}^{(h)} = \text{LeakyReLU}\Big( \langle W_h x_i, a_h^{(src)} \rangle + \langle W_h x_j, a_h^{(dst)} \rangle + \log(w_{ij} + \varepsilon) \Big)$$

$$\alpha_{ij}^{(h)} = \text{softmax}_{k \in \mathcal{N}(j)} \left( \frac{e_{kj}^{(h)}}{T} \right)$$

$$h_j^{(h)} = \sum_{i \in \mathcal{N}(j)} \alpha_{ij}^{(h)} W_h x_i$$

Concatenate heads to obtain layer output;

Stack two cycle-aware attention layers to obtain node embeddings $H \in \mathbb{R}^{N \times d}$;

**Soft community assignments:**

$$Q = \text{softmax}(H)$$

**Loss function:**

$$\mathcal{L} = \lambda_{\text{mod}} \mathcal{L}_{\text{mod}}(Q, E, w) + \lambda_{\text{lap}} \mathcal{L}_{\text{lap}}(H, E, w) + \lambda_{\text{con}} \mathcal{L}_{\text{con}}(H, E, w) + \lambda_{\text{orth}} \mathcal{L}_{\text{orth}}(Q)$$

$$\mathcal{L}_{\text{lap}} = \sum_{(i,j)} w_{ij} \|h_i - h_j\|^2, \quad \mathcal{L}_{\text{orth}} = \|S^\top S - I\|_F, \quad S = \frac{Q}{\mathbf{1}^\top Q + \varepsilon}, \quad \varepsilon = 10^{-9}$$

Optimize $\mathcal{L}$ via gradient descent;

**Final segmentation (hard partition):**

$$\hat{y} \leftarrow \text{KMeans}(H, K)$$

**return** $\hat{y}$

---

Table 8: RCN ablation results on Cora using node features vs. one-hot inputs and RNBRW weighting vs unweighted. Results are mean $\pm$ std over 5 seeds.

| Model | Structural $k$ | | | Ground-truth $k$ | | |
| --- | --- | --- | --- | --- | --- | --- |
| | NMI | ARI | Sil. | NMI | ARI | Sil. |
| Unweighted Features | $0.18 \pm 0.06$ | $0.26 \pm 0.041$ | $0.10 \pm 0.05$ | $0.04 \pm 0.06$ | $0.10 \pm 0.09$ | $0.50 \pm 0.32$ |
| Weighted Features | $0.19 \pm 0.06$ | $0.30 \pm 0.06$ | $0.27 \pm 0.03$ | $0.17 \pm 0.05$ | $0.28 \pm 0.05$ | $0.31 \pm 0.03$ |
| Unweighted One-hot | $0.00 \pm 0.00$ | $0.04 \pm 0.00$ | $0.83 \pm 0.02$ | $-0.01 \pm 0.01$ | $0.03 \pm 0.01$ | $0.85 \pm 0.01$ |
| Weighted One-hot | $0.27 \pm 0.02$ | $0.38 \pm 0.01$ | $0.52 \pm 0.01$ | $0.26 \pm 0.04$ | $0.36 \pm 0.03$ | $0.38 \pm 0.04$ |

| Varied term | $m$ | ONMI | Silhouette | DB | CH | PredComms$_{\geq 2}$ |
|---|---|---|---|---|---|---|
| $\lambda_{\mathrm{con}}$ | 0 | 0.331±0.007 | 0.052±0.002 | 0.587±0.002 | 45.2±0.6 | 459±10 |
| $\lambda_{\mathrm{con}}$ | 0.5 | 0.346±0.006 | 0.110±0.001 | 0.503±0.001 | 93.8±4.4 | 559±8 |
| $\lambda_{\mathrm{con}}$ | 1 | 0.342±0.006 | 0.108±0.006 | 0.499±0.005 | 93.4±5.2 | 558±1 |
| $\lambda_{\mathrm{con}}$ | 2 | 0.339±0.003 | 0.109±0.005 | 0.492±0.001 | 97.2±3.1 | 552±9 |
| $\lambda_{\mathrm{lap}}$ | 0 | 0.341±0.003 | 0.108±0.001 | 0.498±0.006 | 94.8±3.0 | 558±2 |
| $\lambda_{\mathrm{lap}}$ | 0.5 | 0.342±0.008 | 0.109±0.001 | 0.500±0.005 | 94.6±2.9 | 558±1 |
| $\lambda_{\mathrm{lap}}$ | 1 | 0.342±0.006 | 0.108±0.006 | 0.499±0.005 | 93.4±5.2 | 558±1 |
| $\lambda_{\mathrm{lap}}$ | 2 | 0.340±0.006 | 0.109±0.004 | 0.497±0.007 | 92.8±3.2 | 553±5 |
| $\lambda_{\mathrm{mod}}$ | 0 | 0.343±0.007 | 0.110±0.002 | 0.500±0.006 | 94.4±2.1 | 557±1 |
| $\lambda_{\mathrm{mod}}$ | 0.5 | 0.343±0.008 | 0.109±0.003 | 0.498±0.007 | 94.1±2.8 | 556±7 |
| $\lambda_{\mathrm{mod}}$ | 1 | 0.342±0.006 | 0.108±0.006 | 0.499±0.005 | 93.4±5.2 | 558±1 |
| $\lambda_{\mathrm{mod}}$ | 2 | 0.339±0.005 | 0.108±0.006 | 0.490±0.005 | 90.6±4.5 | 553±8 |
| $\lambda_{\mathrm{orth}}$ | 0 | 0.339±0.008 | 0.107±0.002 | 0.502±0.008 | 94.9±3.6 | 559±1 |
| $\lambda_{\mathrm{orth}}$ | 0.5 | 0.340±0.008 | 0.109±0.003 | 0.499±0.002 | 94.5±1.9 | 555±3 |
| $\lambda_{\mathrm{orth}}$ | 1 | 0.342±0.006 | 0.108±0.006 | 0.499±0.005 | 93.4±5.2 | 558±1 |
| $\lambda_{\mathrm{orth}}$ | 2 | 0.339±0.007 | 0.108±0.004 | 0.497±0.009 | 93.1±2.3 | 555±5 |

Table 9: One-factor-at-a-time sensitivity of loss weights on Complex Portal PPI (fixed $K = 793$ and overlap evaluation $r = 2$). Each row multiplies one loss weight by $m \in \{0, 0.5, 1, 2\}$ while holding the remaining weights at their base values. Values are mean±std over two seeds (120, 42). PredComms$_{\geq 2}$ counts predicted communities that survive the size $\geq 2$ filter used in ONMI.

Table 10: Metrics for original and rewired graphs in the cycle-breaking RCN ablation

| Graph Version | Avg. Clustering Coefficient | Triangle Count | Swap_mult |
|---|---|---|---|
| Original | 0.1322 | 13,151,814 | 10 |
| Rewired | 0.0798 | 8,817,824 | 10 |

