# OpenReview forum: "RECYCLE NET: CYCLE-AWARE, FEATURE-FREE GNN FOR COMMUNITY DETECTION"
_ICLR.cc/2026/Workshop/GRaM — ICLR 2026 Workshop GRaM Poster_

### Official Review · Reviewer_8gfU · 2026-02-11
**Cycle-aware GNN with promising direction but significant methodological and evaluation concerns**

**Rating:** 5
**Confidence:** 3

**Review:**

Summary. The paper proposes ReCycle Net (RCN), a feature-free GNN for unsupervised community detection that integrates Renewal Non-Backtracking Random Walk (RNBRW) edge weights into a GAT backbone. The RNBRW weights are injected as additive bias terms in the pre-softmax attention logits, and the model is trained with a four-term unsupervised loss (modularity, Laplacian smoothness, contrastive, orthogonality). Evaluation is conducted on Karate Club, PolBooks, Cora, Facebook, and a Complex Portal PPI graph, with overlap-aware ONMI reported for the PPI case.Strengths.The core idea of injecting cycle-aware topological signals into GNN attention is well-motivated, particularly for biological networks where short cycles correspond to protein complexes. The paper correctly identifies that standard message-passing GNNs treat cycles only implicitly through multi-hop aggregation, and that making cycle structure explicit could improve community detection in the right regime. The paper is also honest about scope: it explicitly states that RCN targets cycle-rich graphs and treats Cora as a "stress test" where gains should be limited.The cycle-breaking control experiment on the PPI graph (Section 5) is a good methodological contribution. Constructing a degree-preserving surrogate with disrupted cycles and showing ONMI degradation (0.346 → 0.317) provides direct evidence that RNBRW is leveraging cyclic structure rather than acting as a degree-based reweighting. More papers should include controls of this kind.The lambda sensitivity analysis (Table 3) is thorough and reveals that the contrastive loss is the critical component (removing it causes ONMI to drop from ~0.37 to 0.17), while the other three terms are relatively interchangeable. This is an important finding, though it also raises questions about the paper's narrative (discussed below).The PPI overlap evaluation using ONMI with top-r membership is a reasonable approach to handling inherently overlapping ground truth, and the paper makes a valid point that partition-based metrics can understate performance in this setting.Weaknesses.The most serious concern is that the results on the main benchmarks do not support the paper's claims of superiority. On Karate Club, RCN ties DMoN exactly (NMI 0.84, ARI 0.88) but has lower silhouette (0.81 vs. 0.99). On PolBooks, RCN again ties DMoN on NMI/ARI (0.60/0.67) but again with lower silhouette (0.89 vs. 0.99–1.00). On Cora, RCN underperforms Louvain (NMI 0.38 vs. 0.46) and is comparable to DMoN. The abstract claims "competitive or superior performance" and "especially strong gains on PolBooks (NMI 0.60, ARI 0.67)," but these are identical to DMoN's numbers. The paper's own results show that RCN matches but does not exceed DMoN on the standard benchmarks where both can be evaluated with the same metrics.The silhouette metric is used inconsistently and appears to be selectively emphasized. The paper argues that silhouette should be the "primary evaluation criterion" because benchmark labels are class-oriented rather than community-grounded. However, DMoN achieves silhouette scores of 0.99–1.00 on Karate and PolBooks — dramatically higher than RCN's 0.81 and 0.89. The paper does not acknowledge or explain this gap. Instead, it emphasizes silhouette on Facebook (where there are no ground-truth labels, making it the only available metric) and deemphasizes it on the benchmarks where DMoN dominates. This selective reporting undermines trust in the evaluation.DMoN's near-perfect silhouette scores (0.99–1.00) should raise red flags that the paper does not address. These scores likely indicate that DMoN's embeddings have collapsed into extremely tight, well-separated clusters — which the paper actually describes as "over-collapsing" in the toy graph example. But the paper never reconciles this: if DMoN's high silhouette reflects pathological collapse, why is silhouette presented as a reliable primary metric? And if silhouette is reliable, then DMoN clearly outperforms RCN on it. The paper cannot have it both ways.The four-term loss function has four weighting hyperparameters (λ₁–λ₄) that are "optimized via grid search on each dataset." Combined with the RNBRW simulation budget, the elbow-method k selection, and other hyperparameters, the total tuning surface is substantial. The paper does not report how many configurations were evaluated per dataset or provide sensitivity analysis for any dataset other than PPI. Given that the primary benchmarks (Karate, PolBooks) have only 34 and 105 nodes respectively, there is a real risk that the reported results reflect hyperparameter overfitting to these tiny graphs. Running 5 seeds is insufficient protection if the hyperparameters were selected to maximize performance on these same 5 seeds.The lambda sensitivity analysis (Table 3) reveals a surprising finding that the paper underplays: removing the modularity, Laplacian, or orthogonality terms individually barely affects ONMI (all remain ~0.37), while removing the contrastive term causes catastrophic degradation (0.17). This suggests that three of the four loss components are largely redundant on the PPI benchmark, and the contrastive loss alone is doing most of the work. This undermines the paper's framing of a carefully designed "multi-objective" loss and raises the question: would a GAT with RNBRW-weighted contrastive loss alone match RCN's performance? This critical ablation is missing.The paper does not compare against any overlapping community detection methods on the PPI benchmark, despite this being a key selling point. Methods like BigCLAM, CESNA, or even simple clique percolation would be natural baselines for overlapping protein complex detection. Comparing only against DMoN and vanilla GAT (which are partition-based methods being awkwardly adapted to overlap via top-r membership) creates a weak baseline set.The geometry connection claimed in Section 3.3 is speculative and underdeveloped. The paper draws a parallel between RNBRW-weighted shortest paths and Ollivier-Ricci curvature, but provides no formal or empirical analysis connecting these concepts. The sentence "we do not claim an exact equivalence to a smooth Riemannian metric" appropriately hedges, but then the paper doesn't provide any substantive geometric analysis either. For a GRaM workshop paper, the geometry connection should be more than a paragraph of hand-waving.The "feature-free" framing is both a strength and a weakness. While it is admirable to show that community detection can work without node features, the paper does not compare against a version of RCN that uses available node features (e.g., on Cora, where features exist). This makes it impossible to assess whether the RNBRW signal is complementary to or redundant with feature-based attention.

**Pmlr Suitability:**

No

---

### Official Review · Reviewer_TPpM · 2026-02-23
**Missing evaluations**

**Rating:** 6
**Confidence:** 1

**Review:**

The paper proposes a new approach for community detection in graphs - RCN.
It incorporates Renewal Non-Backtracking Random Walk (RNBRW) method
into GAT.
They evaluate their method on several different graph datasets and show
comparable results to DmoN method and improved results on PPI dataset.
Strengths:
- The paper has clear motivation and intuitive method.
- They show how the core idea (RNBRW) and each term in the loss
function contributes to the method.
Weaknesses:
- Lacking resources and time complexity between the methods.
- Could be interesting to see another GNN model like GIN.
- Could be interesting to see if the ablation study results are consistent between the datasets or each has its own characteristics.

**Pmlr Suitability:**

No

---

### Official Review · Reviewer_FdAi · 2026-02-23
**Slightly above Borderline Paper**

**Rating:** 6
**Confidence:** 3

**Review:**

**Summary** This paper notes that existing Graph Neural Networks  fail to explicitly model higher-order cyclic structures that underlie many real-world communities. THe authors then propose ReCycle Net (RCN), a feature-free, cycle-aware GNN, which integrates a Renewal Non-Backtracking Random Walk (RNBRW) reinforcement into a GAT backbone and is trained with multiple unsupervised objectives.

**Strengths:**
1) The paper picks up on an interesting point: Sensibly incorporating information about cycles is indeed important.
2) The cycle breaking experiments (performance drops when cycle structure is removed) are convincing
3) The fact that the method works without node features or labels is great

**Weaknesses:**
1) Many hyperparameters on small datasets: The multi-objective loss introduces several hyperparamers that require tuning. At the same time, experiments are performed mostly (aside from PPI) on small datasets. Hence it is difficult to determine how much is due to these additionally tuned hyperparameters, not present for other models.
2) The applicability is somewhat narrow. The focus is on cycle heavy graphs and gains shrink when considering Cora.

**Verdict:** Since this is a workshop, I think the merits of the submission outweigh the concerns slightly.

**Pmlr Suitability:**

Yes

---

### Meta-Review · Area_Chair_yFTR · 2026-02-26

**Decision:**

Accept

**Metareview:**

The reviewers agree that the method is well motivated and that the evaluation is sound. However, some claims should be nuanced. The large number of hyperparameters is also a concern given the small graphs used in benchmarks. The authors are strongly encouraged to address these.

**Relevance To Proceedings:**

Yes — suitable for PMLR (long paper)

**Relevance To Workshop:**

Yes — suitable for GRaM

---

### Decision · Program_Chairs · 2026-03-02

Accept (Poster)